

# Bond strength between layering ceramic and zirconia core: influence of various surface conditioning of zirconia

Abdulaziz A. AlHelal[1], Syed Rashid Habib[1], Saleh Alyousef[2], Nawaf Alhamzah[2], Meshal Alqahtani[2] and Abdulaziz Alqahtani[1]

[1] Department of Prosthetic Dental Sciences, College of Dentistry, King Saud University, Riyadh, Saudi Arabia
[2] Dental Internship Program, College of Dentistry, King Saud University, Riyadh, Saudi Arabia

Corresponding author
Syed Rashid Habib,
syhabib@ksu.edu.sa

## ABSTRACT

**Objective**. It is recommended to apply various surface modification techniques to improve the mechanical and chemical interaction between zirconia core and veneering ceramic. Therefore, the objective of this *in-vitro* study was to evaluate and compare the effects of various surface treatments over the zirconia core on the shear bond strength (SBS) between layering ceramic and zirconia core.

**Methods**. A total of 90 zirconium oxide (Cercon, Degudent GmbH, Hanau-Wolfgang, Germany) specimens were prepared with standardized dimensions (10 mm × 10 mm × 5 mm). The specimens were randomly divided into six groups of 15 specimens each (Control group-no surface treatment; Sandblasting with 25 $\mu$m $Al_2O_3$; Sandblasting with 50 $\mu$m $Al_2O_3$; Sandblasting with 110 $\mu$m $Al_2O_3$; Etching with hydrofluoric acid (9.5%); Surface treatment using Zirface (nano layer etching technology for enhanced bonding strength of zirconia). All specimens were veneered with fluorapatite glass-ceramic (5 mm × 5 mm × 5 mm) using a custom silicone mold, sintered in a calibrated porcelain furnace, and verified with a digital caliper. Following veneering, the samples were embedded in self-curing acrylic resin within polyvinyl chloride molds and immersed in distilled water at 37 °C for 24 hours to simulate intraoral conditions. Using universal testing machine, specimens were tested for SBS. Analysis of variance (ANOVA) and Tukey's-test were used for statistical analysis ($P \leq 0.05$). Surface topography of the surfaces treated and after debonding were examined using scanning electron microscope (SEM) and failure mode of different groups was also noted.

**Results**. The application of Zirface to the zirconia core had the highest mean SBS value ($26.56 \pm 1.25$ MPa), followed by the hydrofluoric acid only group ($23.42 \pm 0.94$ MPa). The control group with no surface treatment showed the least SBS values. While the groups with surface treatments of 25 and 50 aluminum oxide ($Al_2O_3$) sandblasting showed SBS values of $17.16 \pm 0.48$ and $18.06 \pm 0.89$, respectively, with no significant difference ($p > 0.05$) between the SBS values of the two groups. However, the sandblasting with 110 $Al_2O_3$ particles over the zirconia core surface showed higher SBS values ($20.53 \pm 1.14$) as compared to the 25 and 50 $Al_2O_3$ particles.

**Conclusion**. The study found that the shear bond strength between zirconia and veneering ceramic is significantly influenced by the surface treatment used, with nano-layer etching and hydrofluoric acid etching enhancing adhesion, and larger alumina particles enhancing bonding durability.

## INTRODUCTION

Over the years, the concept of 'esthetic' in dentistry has evolved considerably, shaping both patient expectations and treatment approaches. As the demand for highly natural and lifelike restorations has grown, porcelain-fused-to-metal (PFM) crowns—once regarded as the gold standard for fixed partial dentures (FPDs) for nearly four decades—are increasingly being replaced by all-ceramic crowns. These newer restorations provide superior esthetic outcomes and more closely mimic the optical properties of natural teeth while still fulfilling the primary objective of repairing damaged tooth structures (*D'Souza et al., 2025*; *Newaskar et al., 2022*; *Gallucci et al., 2011*; *Christensen, 2009*). This dependable option offers mechanical qualities including high flexural and shear bond strength (SBS) in addition to cosmetic qualities that resemble those of genuine teeth. The development of a new non-metal restoration is justified in light of the growing demand for aesthetic restorations in recent years and the debatable toxicity of certain dental alloys (*Abrisham et al., 2017*). Additionally, metal–ceramic restorations have been linked to certain clinical drawbacks, including the induction of a toxic or allergic reaction in the surrounding tissues, a slight grey discoloration in the surrounding gingiva due to the metal component, a decrease in the vitality of the abutment teeth, and an increased risk of abutment fractures. Instead of porcelain-fused-to-metal and full metal restorations, tooth-colored restorations have emerged as a result of the growing aesthetic demands of patients and physicians, which prompted a search for the ultimate esthetic restorative material (*Zhang & Kelly, 2017*; *Warreth & Elkareimi, 2020*).

Recent advancements in all-ceramic materials, including translucent, ultra-translucent, and super-translucent zirconium blocks, are being introduced at a rapid pace due to their potential to enhance aesthetics, biocompatibility, and long-term stability. Currently, a wide range of techniques are available for the fabrication of all-ceramic fixed partial prostheses (*Alqutaibi et al., 2022*; *Alrabeah, Al-Sowygh & Almarshedy, 2024*; *Lopez-Suarez et al., 2025*). Since zirconia ceramics are now available in dentistry, all-ceramic restorations are increasingly popular and come in a greater range of designs and applications. Zirconia is often the preferred substructure due to its exceptional mechanical properties, which remain consistent regardless of the size or limitations of the restoration site. In addition, zirconia-based materials are considered among the most reliable options for a wide range of therapeutic indications, thanks to the precision and versatility offered by CAD/CAM technology (*Alqutaibi et al., 2022*; *Vijan, 2024*). However, the translucency of zirconia restorations is lower than that of other ceramic materials because of the high crystalline concentration, thick grain, and big grain diameters, but nevertheless, for posterior single crowns or FPDs, it is a viable solution (*Nakai et al., 2025*; *Yousry et al., 2024*).

The use of ceramics has been limited due to its intrinsic brittleness and other properties, especially in posterior teeth. Ceramic chipping and fracture in bilayered restorations like

metal-ceramic and layered-zirconia can be distressing for patients and doctors alike, and they should be addressed seriously. Even while repairing the fracture with composite material or simply polishing the rough edges may be sufficient in certain cases, patients may need to replace the complete restoration (*Aoki et al., 2024*). Numerous factors have been suggested as reasons for ceramic chipping and fracture, such as the layering porcelain's mechanical deficiency, the framework's insufficient support of the layering porcelain, an imbalance in the coefficient of thermal expansion, and unfavorable shear forces between the zirconia framework and the layering materials (*Alayad, 2019*). Some possible causes of chipping of the ceramic layer overlying the zirconia core include inadequate substrate support, excessive load from premature contacts, tensile stress created during cooling after firing, and inadequate bond strength and excessive tensile stress resulting from a mismatch in the coefficient of thermal expansion. This is especially true when a significant thermal gradient forms through the layered system upon rapid cooling (*Brandeburski & Della Bona, 2020*).

Bond strength is influenced by several factors, including the surface treatment of the framework, flaws at the interface between the core and veneer, residual stress from the heat mismatch between the core and porcelain veneer, surface wetting characteristics, and interactions between substances at the interface (*Youssef, Abdelkader & Aly, 2023*). While treatments with 4%–10% hydrofluoric acid (HFC) have no impact on zirconia core at ambient temperature, HF acid etching typically improves the general resin bonding of traditional ceramics due to its extremely dense crystalline and silica-free structure (*Jin et al., 2022*). In addition, it is recommended to apply various surface modification techniques to improve the mechanical and chemical interaction between zirconia core and veneering ceramic, such as varying particle sizes and oxides in sandblasting (*Mohit et al., 2022*; *Karthigeyan et al., 2019*). Despite the growing clinical use of zirconia-based restorations, there is still no consensus on the most effective surface treatment to achieve durable bonding between zirconia cores and veneering ceramics. The literature reports conflicting results: a systematic review noted substantial heterogeneity in outcomes across various pretreatment methods, with no single protocol emerging as definitively superior (*Comino-Garayoa et al., 2021*). Likewise, another study explicitly stated that "it has not been determined which of these treatments produces the highest bond strength" (*Kim et al., 2011*). Therefore, this study was conducted to evaluate and compare the effects of various surface treatments on the zirconia core in relation to the SBS between the layering ceramic and zirconia. The surface treatments included sandblasting with 25/50/110 $\mu$m aluminium oxide ($Al_2O_3$), application of 10% HFC only, and application of Zirface (nano-layer etching technology for enhanced bonding strength of zirconia). The null hypothesis was that there would be no statistically significant difference in the SBS between the layering ceramic and zirconia core surfaces subjected to different surface treatments, including sandblasting with 25/50/110 $\mu$m $Al_2O_3$, application of 10% HFC only, and with the application of Zirface. Furthermore, the surface topography of the treated specimens was evaluated using a SEM both before and after debonding.

## MATERIALS AND METHODS

### Ethical approval

This *in-vitro* experimental study was conducted in the College of Dentistry Research Center (CDRC) at King Saud University's College of Dentistry in Riyadh, Saudi Arabia. Ethical permission was given by the CDRC (# lR 0513).

### Sample size calculation

The sample size was generated with the G*Power software (version 3.1.9.3, Heinrich-Heine-Universität Düsseldorf, Germany). Based on an alpha level of 0.05, a statistical power of 0.9, and a medium effect size of 0.45, the required total sample size was determined to be 90, evenly distributed over six groups ($n = 15$ each group).

### Specimen preparation and surface treatments

In this study, ninety zirconium oxide specimens (Cercon, DeguDent GmbH, Hanau-Wolfgang, Germany) were prepared, each measuring 10 mm $\times$10 mm $\times$5 mm. The specimens were sectioned from pre-sintered zirconia blocks using an Isomet® precision diamond saw (Isomet 1000, Buehler Ltd., Lake Bluff, IL, USA) under continuous water irrigation to prevent overheating and minimize microcrack formation. Following sectioning, the specimens were sintered in a high-temperature furnace (Programat® S1, Ivoclar Vivadent, Schaan, Liechtenstein) at 1,450 °C for 2 h following the manufacturer's sintering protocol to achieve full densification. The dimensions of all specimens were verified using a digital caliper (Mitutoyo Corp., Kawasaki, Japan) with an accuracy of $\pm0.01$ mm. Surface cleaning was performed to eliminate debris, contaminants, and residues that could interfere with subsequent bonding procedures. Each specimen was ultrasonically cleaned in distilled water for 5 min, followed by immersion in 70% ethanol for 10 min to remove organic impurities and ensure surface decontamination. The specimens were then rinsed thoroughly with distilled water and air-dried using oil-free compressed air. This standardized cleaning protocol ensured a uniform and uncontaminated surface prior to surface treatment and bonding procedures.

The specimens were then randomly divided into six groups, with 15 specimens assigned to each group, and each group received a different surface treatment. A summary of the materials used in the study is presented in Table 1. Each group was subjected to a distinct surface treatment, as outlined below;

1. **Control group:** No surface treatment.
2. **Sandblasting with 25 $\mu$m Al$_2$O$_3$:** The specimens were sandblasted with 25 $\mu$m Al$_2$O$_3$ for 15 s at a pressure of 2.8 atm, maintaining a nozzle-to-surface distance of 10 mm.
3. **Sandblasting with 50 $\mu$m Al$_2$O$_3$:** The specimens were sandblasted with 50 $\mu$m Al$_2$O$_3$ for 15 s at a pressure of 2.8 atm, maintaining a nozzle-to-surface distance of 10 mm.
4. **Sandblasting with 110 $\mu$m Al$_2$O$_3$:** The specimens were sandblasted with 110 $\mu$m Al$_2$O$_3$ for 15 s at a pressure of 2.8 atm, maintaining a nozzle-to-surface distance of 10 mm.
5. **Etching with HFC:** The specimens were etched with 9.5% HFC for 20 s, thoroughly rinsed with water for another 20 s, and then air-dried.

**Table 1  Details of the materials used and tested in the study.**

| Materials | Brand | Manufacturer | Chemical composition | Lot No. |
|---|---|---|---|---|
| Zirconia core | DEGUDENT Cercon-Base | Degudent GmbH, hanau wolfgang, Germany | Zirconium dioxide ($ZrO_2$) Yttria ($Y_2O_3$) Alumina ($Al_2O_3$) | 18007661 |
| Layering ceramic | IPS e.max® Ceram | Ivoclar Vivadent, Schaan, Liechtenstein | Fluorapatite glass-ceramic | R60559 |
| Zirface | Zirface Z-Etch | DMAX Dental America Corp Korea | Zirconium dioxide Water Carbon | 240502ZF |
| Hydrofluoric acid | Porcelain Etchant 9.5% Gel | Bisco Porcelain Etchant; BISCO Inc., Schaumburg, IL | Hydrofluoric acid (HF) Water ($H_2O$) | 1600005709 |
| Sandblasting 110 | Aluminum Oxide (110 μm) | 3M™ ESPE, St. Paul, Minnesota, USA | Synthetic Amorphous silica, Fumed | 15831005 |
| Sandblasting 50 | Aluminum Oxide (50 μm) | 3M™ ESPE, St. Paul, Minnesota, USA | Synthetic Amorphous silica, Fumed | 2567108 |
| Sandblasting 25 | Aluminum Oxide (25 μm) | 3M™ ESPE, St. Paul, Minnesota, USA | Synthetic Amorphous silica, Fumed | 15851005 |

6. **Surface treatment using Zirface:** The specimens were air-cleaned, and after thoroughly shaking the Zirface container, a thin, uniform layer of Zirface was applied to their surfaces using a small brush. Subsequently, the specimens were sintered at 1,530 °C for 10 min (Fig. 1).

## Evaluation of surface roughness

Surface roughness (Ra) was evaluated on representative specimens from control group with no surface treatment and for each treatment group using a 3D optical profilometer (Contour-GT-X®, Bruker Nano Surfaces Division, San Jose, CA, USA) following the surface treatments. Three measurements were recorded for each specimen, and the average of these values were calculated and reported as the final roughness value for that respective group.

## Veneering and aging

After completing the surface treatments, all specimens were manually veneered with a layering ceramic measuring 5 mm × 5 mm × 5 mm and subsequently sintered in a ceramic furnace. Prior to veneering, the zirconia specimens were thoroughly cleaned using an ultrasonic bath. A custom-made detachable silicone mold was employed to apply the veneering porcelain onto the treated surfaces.

For the base dentin wash bake, base dentin powder (Vita Zahnfabrik, Bad Säckingen, Germany) was mixed with a modeling liquid to form a thin slurry, which was lightly applied to the zirconia specimens and fired in a ceramic furnace (Vita Zahnfabrik, Bad Säckingen,

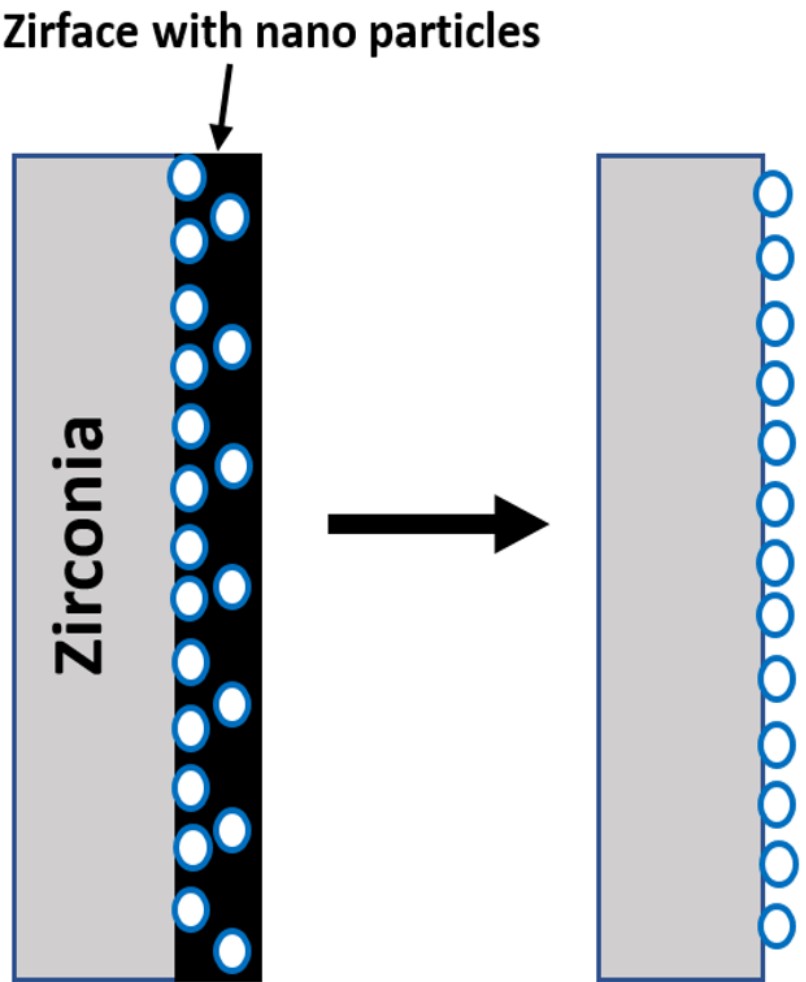

**Before Sintering**     **After Sintering**

**Figure 1** **Schematic illustration of Zirface surface treatment on zirconia.** Zirface is applied before sintering, coating the surface with nano-sized zirconia ($ZrO_2$) particles. During sintering, carbon and other components are eliminated, leaving firmly attached $ZrO_2$ particles that create a highly porous and roughened surface.

Germany) according to the manufacturer's guidelines. The fluorapatite glass-ceramic (IPS e.max® Ceram, Ivoclar Vivadent, Schaan, Liechtenstein) was then prepared by mixing the ceramic powder with the recommended liquid, following the manufacturer's instructions, and applied over the zirconia blocks in dimensions of 5 mm × 5 mm × 5 mm. Excess liquid was carefully removed using tissue paper before firing the specimens in a calibrated porcelain furnace (Esgaia, J. Morita Mfg Corp, Kyoto, Japan) as per the manufacturer's recommendations. Finally, the veneered specimens were finished, and the dimensions of the ceramic layers were verified using a digital caliper.

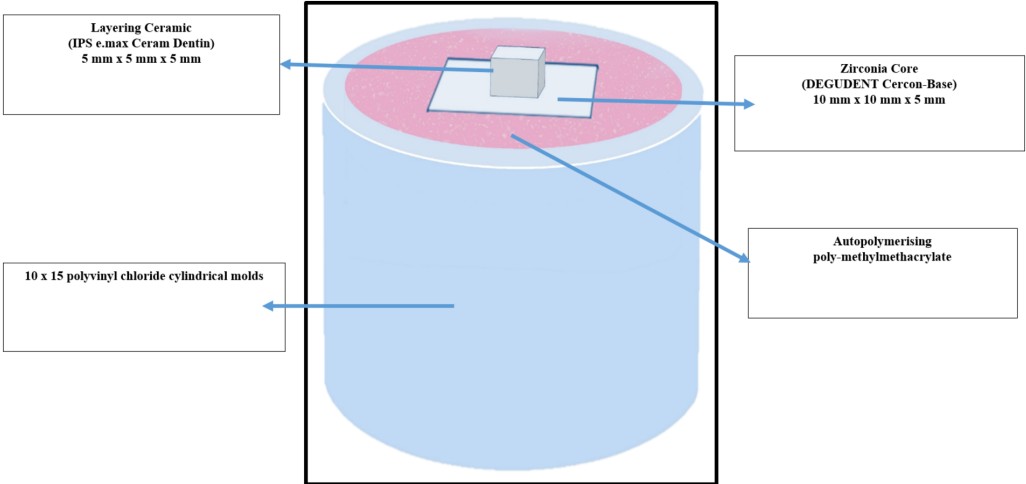

**Figure 2** Schematic diagram showing specimen size and test design.

Following the fabrication, all specimens were embedded in self-curing acrylic resin using polyvinyl chloride (PVC) cylindrical molds to facilitate standardized handling and testing (Fig. 2). The embedding process ensured stability during subsequent procedures and minimized the risk of specimen displacement or damage. To closely replicate the intraoral environment, the embedded specimens from all experimental groups were immersed in distilled water and stored at 37 °C for 24 h. This step aimed to achieve water saturation of the ceramic–resin interface and to simulate the moisture conditions typically encountered in the oral cavity.

After storage, artificial aging was performed using a thermocycling machine (SD Mechatronik Thermocycler; Huber, Berching, Germany). The specimens underwent 6000 thermal cycles between 5 °C and 55 °C, with a dwell time of 30 s at each temperature and a 5-second transfer interval, to mimic the temperature variations experienced in the oral cavity during food and beverage consumption. Each cycle involved a dwell time of 30 s in each water bath, ensuring adequate exposure to both extreme temperatures, followed by a 5-second transfer interval between baths. This thermocycling protocol was selected based on widely accepted *in vitro* standards (*Kim et al., 2023*) to simulate approximately five years of intra oral service for ceramic restorations, thereby providing a more realistic assessment of the bond performance under long-term functional conditions.

## Shear bond strength testing

The molds were then locked in the testing machine (Fig. 3). Using a universal testing machine (Instron 5965 Dual Column Tabletop Testing System, Norwood, MA, USA) the specimens were tested for shear bond strength by applying vertical force for debonding of the layering porcelain from the zirconium core specimens for all the group specimens at a constant crosshead speed of 1.0 mm/min until failure occurred, consistent with widely used protocols in studies of porcelain-to-metal/zirconia, resin-cement, and orthodontic

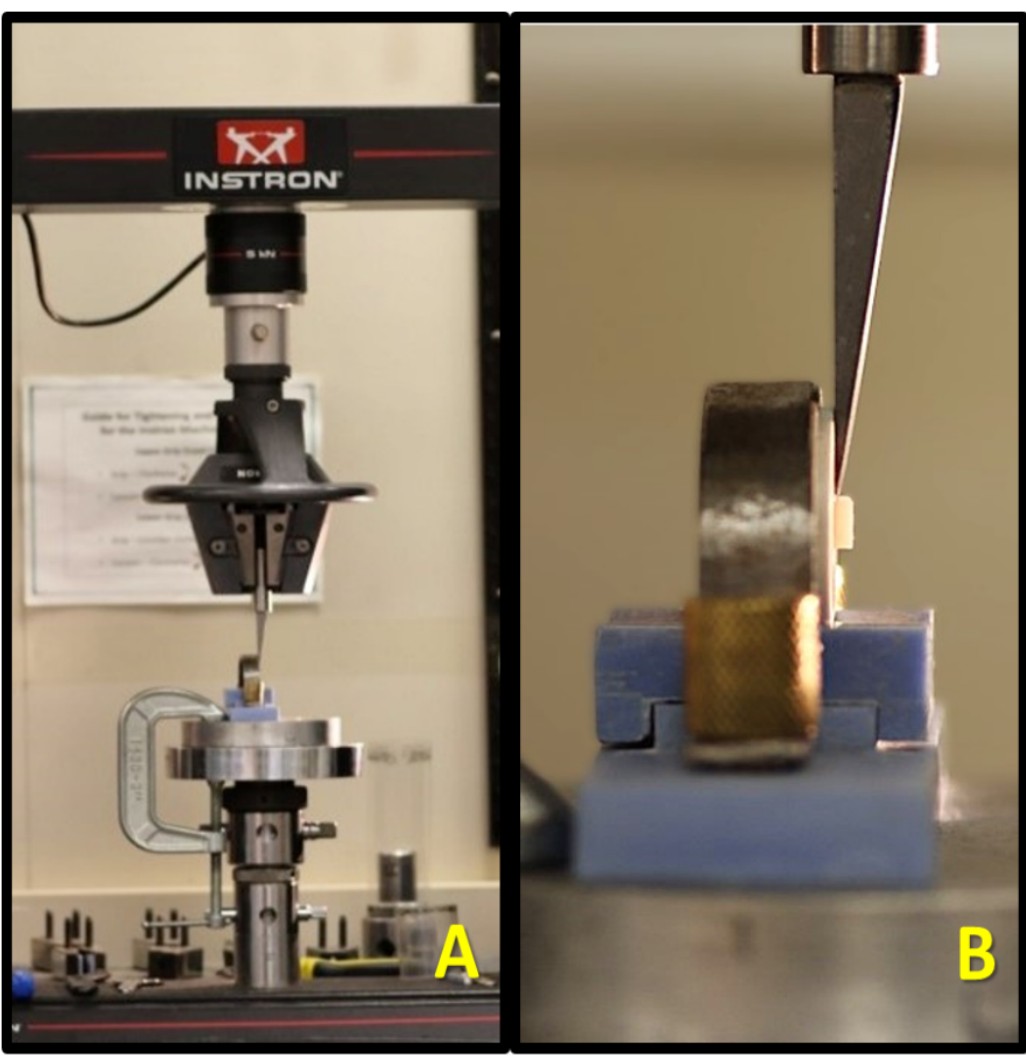

**Figure 3** (A) Specimens locked in the universal testing machine. (B) Close up of the specimens locked in the universal testing machine for the shear bond test (SBS).

bracket bonding (*Limpuangthip, Surintanasarn & Vitavaspan, 2023*). The SBS at the failure of the layered ceramic was noted for each specimen and mean value was calculated for each group in mega pascals (MPa).

## SEM analysis

The surface topography of the zirconia cores was thoroughly examined at three different stages using a scanning electron microscope (SEM) (Fig. 4): (i) before the application of any surface treatments to establish the baseline surface morphology, (ii) after the completion of the respective surface treatments to evaluate the induced microstructural alterations, and (iii) following the debonding of the veneering ceramic to assess the nature and extent of surface changes resulting from the failure. Additionally, the failure modes observed in

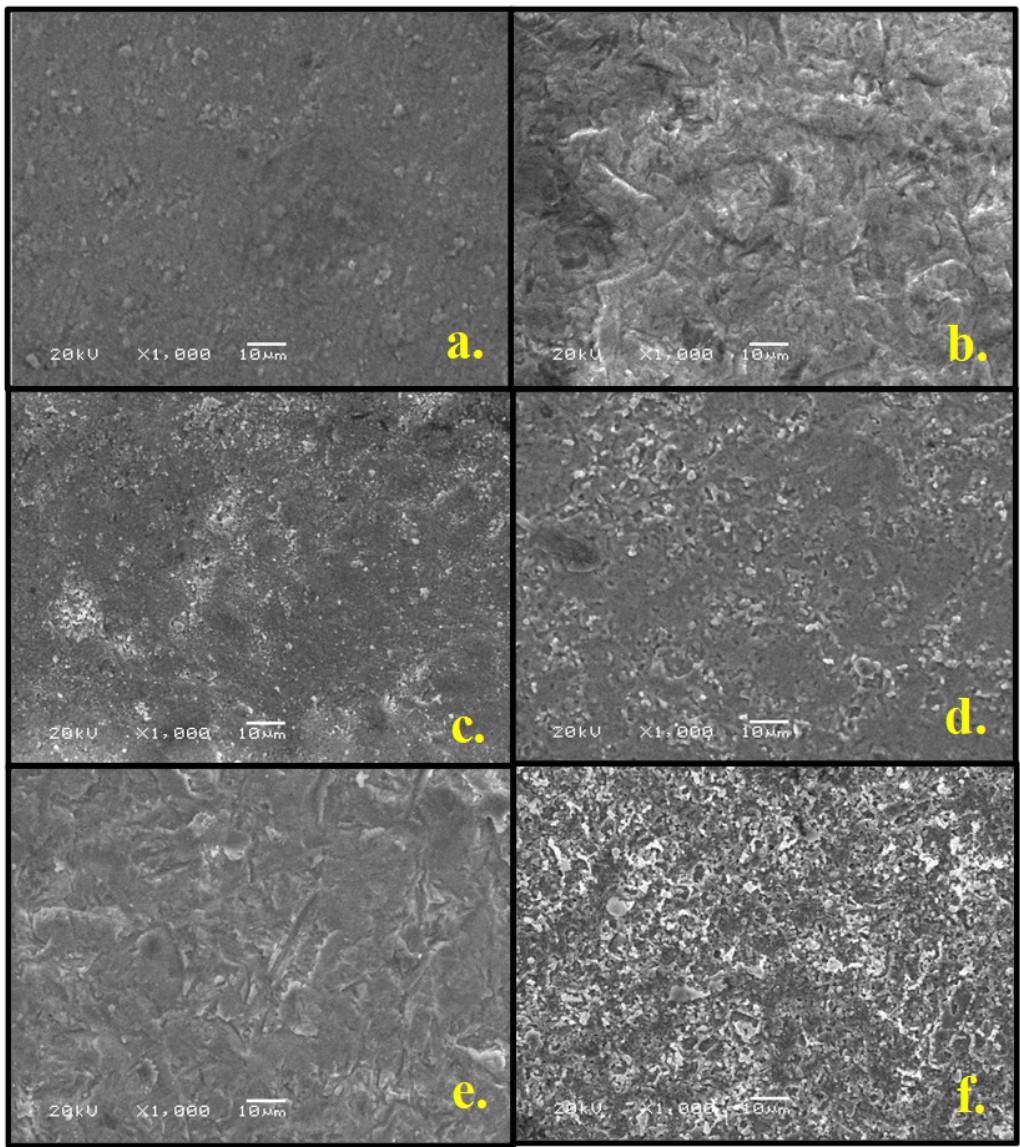

**Figure 4 Scanning electron microscopic images (1,000X magnification) of the test groups after surface treatments and before application of layering ceramic.** (A) Control group; (B) sandblasting with 25 μm alumina; (C) sandblasting with 50 μm alumina; (D) sandblasting with 110 μm alumina; (E) application of hydroflouric acid; (F) application of Zirface.

the different experimental groups were carefully analyzed and documented to correlate the surface characteristics with the bonding performance.

## Statistical analysis

Statistical analysis was performed using SPSS software (version 26.0; IBM Corp., Chicago, IL, USA). The mean SBS values among the experimental groups were compared using one-way ANOVA, followed by Tukey's post hoc test for pairwise comparisons. A confidence level of 95% was applied, with statistical significance defined as $p < 0.05$.

**Table 2  Mean and standard deviation of shear bond strength values (MPa) for the six experimental groups.**

|  | Groups | Mean | Std. deviation | Std. error | 95% Confidence interval for mean | | Minimum | Maximum | *ANOVA |
|---|---|---|---|---|---|---|---|---|---|
|  |  |  |  |  | Lower bound | Upper bound |  |  |  |
| I | No Treatment | 14.56 | 1.023 | .26 | 13.99 | 15.12 | 12.78 | 16.31 |  |
| II | SandBlasting25 | 17.16 | .48 | .12 | 16.89 | 17.43 | 16.48 | 18.31 |  |
| III | SandBlasting50 | 18.06 | .89 | .23 | 17.57 | 18.56 | 16.61 | 19.64 | 0.000 |
| IV | SandBlasting110 | 20.53 | 1.14 | .29 | 19.90 | 21.16 | 18.44 | 22.54 |  |
| V | HFC | 23.42 | .94 | .24 | 22.90 | 23.94 | 21.64 | 25.03 |  |
| VI | Zirface | 26.56 | 1.25 | .32 | 25.86 | 27.25 | 24.72 | 28.52 |  |
|  | Total | 20.05 | 4.14 | .43 | 19.18 | 20.92 | 12.78 | 28.52 |  |

Notes.

*P value was significant at $P < 0.05$.

## RESULTS

The results of this study demonstrated that the type of surface treatment applied to the zirconium oxide ceramic core had a significant impact on the SBS between the layering ceramic and the zirconia core. Among all the groups, the application of Zirface yielded the highest mean SBS value ($26.56 \pm 1.25$ MPa), followed by the HFC etching group ($23.42 \pm 0.94$ MPa). In contrast, the control group without any surface treatment exhibited the lowest SBS values (Table 2).

For the sandblasting groups, treatments using 25µm and 50µm $Al_2O_3$ particles resulted in mean SBS values of $17.16 \pm 0.48$ MPa and $18.06 \pm 0.89$ MPa, respectively, with no statistically significant difference between them ($p > 0.05$). However, sandblasting with 110 µm $Al_2O_3$ particles produced a significantly higher SBS value ($20.53 \pm 1.14$ MPa) compared to the 25 µm and 50 µm groups.

Among all the tested surface treatments, the application of Zirface to the zirconia core achieved the highest mean SBS, which was significantly greater than that of all other groups ($p < 0.05$), showing only a slight, non-significant difference compared to the HFC group. Both the Zirface and HFC treatments resulted in a marked enhancement of SBS compared to the other surface treatments. In contrast, sandblasting with 25 µm $Al_2O_3$ particles produced the lowest mean SBS value, which was significantly lower than most groups ($p < 0.05$), except when compared with the 50 µm $Al_2O_3$ sandblasting group, where no significant difference was observed (Table 3).

Table 4 displays the surface roughness (Ra) values for each of the test groups. The Zirface group had the highest values (Ra = 0.533), followed by the HFC group (Ra = 0.450). The control group had the lowest Ra values (Ra = 0.133), as expected. The sandblasted groups showed an increase in surface roughness with increase in the size of the $Al_2O_3$ particles.

### Fracture analysis

The fractured surfaces of the specimens were examined using a scanning electron microscope (SEM) (JEOL, JSM-6360LV, Tokyo, Japan) to identify the failure modes. Failures occurring at the interface between the zirconia core and the veneering ceramic

**Table 3   SBS values (Mpa) within the groups.** *Post hoc* tests multiple comparisons SBS Tukey HSD.

| Groups | Compared to; | Mean difference | Sig. | 95% Confidence interval | |
|---|---|---|---|---|---|
| | | | | Lower bound | Upper bound |
| NoTreatment | SandBlasting25 | −2.60410[*] | .000 | −3.6558 | −1.5524 |
| | SandBlasting50 | −3.50607[*] | .000 | −4.5578 | −2.4543 |
| | SandBlasting110 | −5.97482[*] | .000 | −7.0265 | −4.9231 |
| | HFC | −8.86303[*] | .000 | −9.9148 | −7.8113 |
| | Zirface | −12.00150[*] | .000 | −13.0532 | −10.9498 |
| SandBlasting25 | NoTreatment | 2.60410[*] | .000 | 1.5524 | 3.6558 |
| | SandBlasting50 | −.90196 | .135 | −1.9537 | .1498 |
| | SandBlasting110 | −3.37072[*] | .000 | −4.4224 | −2.3190 |
| | HFC | −6.25893[*] | .000 | −7.3106 | −5.2072 |
| | Zirface | −9.39739[*] | .000 | −10.4491 | −8.3457 |
| SandBlasting50 | NoTreatment | 3.50607[*] | .000 | 2.4543 | 4.5578 |
| | SandBlasting25 | .90196 | .135 | −.1498 | 1.9537 |
| | SandBlasting110 | −2.46876[*] | .000 | −3.5205 | −1.4170 |
| | HFC | −5.35697[*] | .000 | −6.4087 | −4.3052 |
| | Zirface | −8.49543[*] | .000 | −9.5471 | −7.4437 |
| SandBlasting110 | NoTreatment | 5.97482[*] | .000 | 4.9231 | 7.0265 |
| | SandBlasting25 | 3.37072[*] | .000 | 2.3190 | 4.4224 |
| | SandBlasting50 | 2.46876[*] | .000 | 1.4170 | 3.5205 |
| | HFC | −2.88821[*] | .000 | −3.9399 | −1.8365 |
| | Zirface | −6.02667[*] | .000 | −7.0784 | −4.9750 |
| HFC | NoTreatment | 8.86303[*] | .000 | 7.8113 | 9.9148 |
| | SandBlasting25 | 6.25893[*] | .000 | 5.2072 | 7.3106 |
| | SandBlasting50 | 5.35697[*] | .000 | 4.3052 | 6.4087 |
| | SandBlasting110 | 2.88821[*] | .000 | 1.8365 | 3.9399 |
| | Zirface | −3.13846[*] | .000 | −4.1902 | −2.0867 |
| Zirface | NoTreatment | 12.00150[*] | .000 | 10.9498 | 13.0532 |
| | SandBlasting25 | 9.39739[*] | .000 | 8.3457 | 10.4491 |
| | SandBlasting50 | 8.49543[*] | .000 | 7.4437 | 9.5471 |
| | SandBlasting110 | 6.02667[*] | .000 | 4.9750 | 7.0784 |
| | HFC | 3.13846[*] | .000 | 2.0867 | 4.1902 |

**Notes.**

*The mean difference is significant at the 0.05 level.

were classified as adhesive failures. When the fracture occurred within the veneering ceramic itself, it was considered a cohesive failure. Mixed failures were identified when remnants of the veneering ceramic remained on the zirconia surface, with both materials visibly exposed. Representative SEM images illustrating each failure type are presented in Fig. 2, while the distribution of failure modes across all groups is summarized in Table 5.

Failure mode of specimens after bond test was mostly adhesive *i.e.,* failure between the zirconia and veneering ceramic rather than the cohesive within the ceramic itself only. For the groups with No-Treatment, Sandblasting-25, Sandblasting-50 and Sandblasting-110 no cohesive failure was observed and almost all or most of the failures modes were of

**Table 4  An account of the profilometric measurements of surface roughness (Ra) obtained from each test group.**

| Groups | [a]Surface roughness (Ra) |
|---|---|
| No treatment | 0.133 |
| Sandblasting 25 | 0.245 |
| Sandblasting 50 | 0.327 |
| Sandblasting 110 | 0.347 |
| HFC | 0.450 |
| Zirface | [b]0.533 |

Notes.
[a]Surface Roughness (Ra): Arithmetic mean of the absolute values of the surface profile deviations from a mean line over a specified sampling length.
[b]Mean surface roughness value for Zirface was significantly higher than the other groups.

**Table 5  Comparison of the type of failure modes between the different surface treatment groups.**

| Surface treatment | Failure modes | | | Total |
|---|---|---|---|---|
| | Adhesive N (%) | Cohesive N (%) | Mix N (%) | |
| No-Treatment | 15 (**100**) | 0 (0) | 0 (0) | 15 |
| SandBlasting-25 | 13 (86.6) | 0 (0) | 2 (13.3) | 15 |
| SandBlasting-50 | 12 (80) | 0 (0) | 3 (20) | 15 |
| SandBlasting-110 | 8 (53.3) | 0 (0) | 7 (46.6) | 15 |
| HFC | 5 (33.3) | 4 (26.6) | 6 (40) | 15 |
| Zirface | 3 (20) | 7 (46.6) | 5 (33.3) | 15 |

adhesive nature. For the HFC group the failure of the specimens comprised of adhesive, cohesive and mixed modes. The Zirface group demonstrated the highest proportion of cohesive failures (46.6%), where fractures occurred within the veneering ceramic itself rather than at the zirconia–ceramic interface. This strongly correlates with the SBS findings, as the Zirface group also recorded the highest mean shear bond strength (26.56 MPa). The improved surface treatment and superior micromechanical interlocking achieved with Zirface significantly enhanced the bond strength between the zirconia core and the veneering ceramic. As a result, the bond became stronger than the veneering ceramic's internal strength, causing fractures to occur within the ceramic layer rather than at the interface, highlighting the effectiveness of the Zirface bonding protocol.

The SEM analysis provided critical insights into the failure modes, confirming that surface treatment significantly affects zirconia–ceramic adhesion. The predominance of adhesive failures in untreated and lightly sandblasted groups reflected weak interfacial bonding. In contrast, the Zirface group demonstrated the highest rate of cohesive failures, aligning with its superior SBS values and indicating robust interfacial strength. The HFC group exhibited mixed failure patterns, supporting the positive influence of HFC etching on chemical adhesion. Moreover, sandblasting with larger alumina particles enhanced micromechanical interlocking, reducing adhesive failures. Collectively, these SEM findings

corroborate the conclusion that the durability of the zirconia–ceramic bond depends on both chemical adhesion and micromechanical retention.

## DISCUSSION

The objective of this study was to assess and compare the influence of various surface treatments on the SBS between a zirconia core and a veneering ceramic. Achieving a durable and reliable bond at this interface is essential for the long-term clinical success of zirconia-based ceramic restorations (*Daou, 2014*). Laboratory-based mechanical testing plays a vital role in guiding material selection, surface treatment protocols, and overall restorative strategies in clinical practice (*Alrabeah et al., 2024*). Standardized test specimens with uniform dimensions and shapes were used in the present study to ensure consistency in the ceramic layering process. Each zirconia specimen underwent a specific surface treatment to investigate its influence on SBS when bonded with a multilayer ceramic. The SBS values were calculated in megapascals (MPa) using an Instron universal testing machine, a widely accepted and reliable method for quantifying bond strength. Additionally, surface topography of the specimens was analyzed using a SEM both before and after the debonding tests. SEM provided valuable insight into the morphological changes resulting from the surface treatments and their correlation with SBS performance. The use of MPa as a unit of measurement offers a practical and standardized means of comparing SBS across different surface treatments, as well as with findings from existing literature. The combined use of universal testing equipment and SEM analysis ensured accurate, repeatable results and enhanced understanding of the bonding interface characteristics.

Several testing methods are available to assess the bond strength between veneering ceramics and core materials, including shear bond, three-point and four-point flexural, tensile, and microtensile tests (*Habib et al., 2021*). Among these, shear bond testing is widely used due to its simplicity; however, tensile and microtensile tests are often recommended as they help minimize the influence of non-uniform interfacial stresses. Despite their widespread application, each testing approach has inherent limitations. In shear bond tests, non-uniform stress distribution is a known concern, and the elastic modulus of the bonding materials can significantly affect the outcomes. Flexural tests, such as three-point and four-point bending, tend to generate maximum tensile stresses near the veneering ceramic surface, which can lead to predictable fracture patterns. Similarly, tensile and microtensile tests present challenges related to specimen geometry and may also produce non-uniform stress distributions at the adhesive interface (*Komine et al., 2012*; *Sirisha et al., 2014*).

Feldspathic porcelain is widely utilized as a veneering material for zirconia frameworks in all-ceramic dental restorations. Over the last decade, various *in vitro* research have been conducted to evaluate the bonding strength between layered porcelain and zirconia ceramics. While metal–ceramic systems have long been established as a dependable solution for fixed prosthodontics and are still regarded the gold standard, there is no clear benchmark for all-ceramic systems (*Talibi, Kaur & Parmar, 2022*). The International Standards Organization (ISO) considers a minimum bond strength of 25MPa between

the metal and the veneering porcelain to be sufficient for metal–ceramic restorations. However, there is currently no recognized criterion for acceptable bond strength for all-ceramic materials (*Daou, 2014*; *Talibi, Kaur & Parmar, 2022*; *Khmaj et al., 2014*).

A thorough understanding of the complex structure of ceramic materials is essential for their effective application by ceramists, dental technicians, and clinicians. The microstructure of ceramics plays a key role in determining their physical and mechanical properties (*Bajraktarova-Valjakova et al., 2018*). Several factors influence the bond strength between the ceramic core and the veneering layer. These include the surface characteristics of the core, which contribute to mechanical interlocking, the potential formation of defects or flaws at the core–veneer interface, wetting behavior, veneer shrinkage during the firing process, and residual stresses arising from mismatches in the coefficients of thermal expansion (CTE) between the two materials (*Youssef, Abdelkader & Aly, 2023*). Achieving a durable bond requires a close match between the CTE values of the core and veneering ceramics to minimize internal stresses. In metal–ceramic systems, it is advised that the veneering ceramic have a slightly lower CTE than the metal framework to provide advantageous residual compressive stresses that improve cracking and chipping resistance. To solve this issue, manufacturers have produced layered ceramics with somewhat lower CTEs over tightly aligned zirconia frameworks. However, utilizing a veneering ceramic with a greater CTE can result in ceramic layer delamination and microcracks. Some research indicated that CTE mismatch had no substantial impact on shear bond strength, and there was no evident association between the two variables (*Youssef, Abdelkader & Aly, 2023*; *Kim et al., 2011*). Despite the lack of consensus on an ideal CTE value, veneering ceramic should have a slightly lower CTE than the zirconia framework to ensure a durable and reliable bond.

This study examined the impact of different surface treatments applied to zirconia cores on their bonding performance with ceramic veneers. The findings demonstrated significant variations in SBS among the tested groups, leading to the rejection of the null hypothesis, which stated that surface treatments would not significantly affect the bond strength between the veneering ceramic and the zirconia core. *Shilpa et al. (2019)* previously reported SBS values ranging from 22 to 31 MPa for commercially available core–veneer ceramic systems. Similarly, another study found SBS values between 20 and 35 MPa (*Kelly et al., 1990*). These values were slightly higher than those observed in the present study, likely due to the use of veneering ceramic cylinders with smaller diameters in those investigations. The reduced bonding surface area would have resulted in greater SBS values when expressed in mega pascals, as the applied shear force was distributed over a smaller area. In contrast, and aligning more closely with our findings, *Silva-Herzog Rivera et al. (2020)* reported lower SBS values ranging from 11 to 15 MPa.

The bond between zirconia and veneering ceramic is achieved through a combination of mechanical interlocking and chemical interactions. The presence of oxides such as $HfO_2$, $Al_2O_3$, $Y_2O_3$, and others within the liner coating may promote stronger chemical bonding at the interface, thereby enhancing the adhesion between the zirconia core and the ceramic veneer (*Youssef, Abdelkader & Aly, 2023*; *Daou, 2014*). The highest shear bond strength in the present investigation was observed in the group VI (Zirface, Nano layer etching

technology for enhanced bonding strength of zirconia). The Zirface treatment coats the zirconia surface with nano-sized zirconia ($ZrO_2$) particles to create a porous, roughened surface. Zirface is applied to the zirconia prior to sintering, and during the sintering process, carbon and other components are removed, leaving only the $ZrO_2$ particles attached to the surface. This results in a significant increase in surface roughness, with studies showing that Zirface-treated specimens exhibit roughness values 2 to 6 times higher than those of control or acid-etched groups. Since the main component of Zirface is nano-ceramic, its application does not alter the mechanical strength or shade of the zirconia restoration. Studies by *López Mollá et al. (2010)* and *Youssef, Abdelkader & Aly (2023)* reported shear bond strength values of 12.70 MPa and 17.98 ± 2.51 MPa, respectively, for zirconia cores treated with a ZirLiner coating and veneered with fluoroapatite-pressed ceramic, which align with the findings of the present study. However, some researchers suggest that the use of liner coatings may, in certain conditions, compromise the bond strength rather than enhance it.

The findings of this study revealed that sandblasting with alumina particles significantly enhanced the bond strength between zirconia and veneering ceramic compared to polished specimens that did not undergo sandblasting. This suggests that sandblasting serves as an effective surface treatment for improving adhesion at the zirconia-ceramic interface (*Su et al., 2015*). These results are in agreement with previous research demonstrating similar improvements in bond strength following sandblasting procedures (*Cheng, Yang & Yan, 2018*; *He et al., 2014*; *Ramakrishnaiah et al., 2016*). In the present study, groups (II, III & IV), sandblasting with different sizes of alumina particles showed improvement in the SBS as compared to the zirconia core surface with no treatment at all. It was interesting to see that the SBS was improved with increase in the size of the particles with sandblasting with 110 μm showed the highest SBS values compare to the sandblasting with the 25 μm or 50 μm particle sizes. In simple terms, when both the application time and pressure remained constant, an increase in powder particle size from 25 μm to 50 μm and subsequently to 110 μm led to a corresponding increase in surface roughness. These results are consistent with those found by a previous study by *Su et al. (2015)*, in which they recommended the use of powder size of 110 μm for dental applications to improve the bonding between zirconia core and layering material.

Among the several chemical surface treatments investigated in the literature to promote bonding, HFC etching emerges as a potential alternative. While other treatments, such as hypophosphorous acid and a 1:1 combination of potassium hydroxide and sodium hydroxide, have shown efficacy in etching zirconia, HFC has the distinct benefit of working quickly at the room temperature (*Ramakrishnaiah et al., 2016*; *Melo et al., 2015*). Hydrofluoric acid is commonly used for etching ceramics due to its ability to dissolve the glassy matrix (*Ramakrishnaiah et al., 2016*). The findings of the present work show that HFC etching of zirconia has resulted in increase in the SBS of the layered ceramic bonded to the zirconia core. According to the study by *Sriamporn et al. (2014)*, applying 9.5% HFC enhances the apparent surface roughness of dental zirconia ceramics by introducing microstructural changes to the surface. Microscopic analysis and surface roughness measurements further confirm the significant alteration of the zirconia surface
following treatment. This could be the reason for the increase in the SBS of the ceramic layer bonded to the zirconia core with HFC surface treatment as compared with airborne-particle abrasion in the present study (*Kim et al., 2022*).

Several limitations of this study should be acknowledged. First, comparing results across studies is challenging due to variations in specimen geometry, design, testing protocols, and fabrication methods. In this study, the specimen geometry did not fully replicate the anatomical configuration of zirconia-based ceramic restorations used clinically. However, the use of standardized specimens ensured a consistent and controlled evaluation of SBS. Second, the *in vitro* nature of the study does not fully capture the complex biological and mechanical environment of the oral cavity. Only shear forces were evaluated, whereas dental restorations *in vivo* are subjected to a combination of mechanical stresses, including tensile, compressive, and flexural forces. Additionally, environmental factors such as thermal fluctuations, pH variations, and cyclic fatigue, which are routinely encountered in the oral environment, were not simulated in this study. Future research should aim to better replicate clinical conditions, incorporating anatomically accurate specimens and more comprehensive mechanical and environmental stressors to provide a more realistic assessment of the bond strength between zirconia cores and veneering ceramics. Consequently, the findings of this study should be interpreted with caution and considered within the context of these limitations. Despite these limitations, the findings of this study suggest that Zirface application is recommended as an effective surface treatment to enhance the SBS between zirconia cores and veneering ceramics. The increased surface roughness due to Zirface facilitates better wettability and ceramic adaptation, leading to a more durable bond. Clinically, this enhancement is highly significant as it addresses one of the most frequent complication of ceramic chipping or delamination. By improving the interfacial integrity, the risk of veneer fracture under functional and parafunctional loads is substantially reduced, ultimately increasing the longevity and reliability of zirconia-based restorations.

## CONCLUSION

Within the limitations of this *in-vitro* study, the findings confirm that the type of surface treatment significantly influences the shear bond strength (SBS) between the zirconia core and veneering ceramic. Among the evaluated techniques, Zirface nano-layer etching achieved the highest improvement in SBS, demonstrating superior micromechanical and chemical bonding potential. Etching with 9.5% hydrofluoric acid (HFA) also produced a substantial increase in bond strength compared to airborne-particle abrasion, indicating its effectiveness as a practical alternative when Zirface is unavailable. Furthermore, sandblasting with larger alumina particles (110 $\mu$m) enhanced SBS more effectively than smaller particle sizes.

### Funding

The research was funded by Ongoing Research Funding Program (ORF-2025-950), King Saud University, Riyadh, Saudi Arabia. The funders had no role in study design, data collection and analysis, decision to publish, or preparation of the manuscript.

### Grant Disclosures

The following grant information was disclosed by the authors:
Ongoing Research Funding Program (ORF-2025-950), King Saud University, Riyadh, Saudi Arabia.

### Competing Interests

The authors declare there are no competing interests.

### Author Contributions

- Abdulaziz A. AlHelal conceived and designed the experiments, authored or reviewed drafts of the article, and approved the final draft.
- Syed Rashid Habib conceived and designed the experiments, analyzed the data, authored or reviewed drafts of the article, and approved the final draft.
- Saleh Alyousef performed the experiments, prepared figures and/or tables, and approved the final draft.
- Nawaf Alhamzah performed the experiments, prepared figures and/or tables, and approved the final draft.
- Meshal Alqahtani performed the experiments, prepared figures and/or tables, and approved the final draft.
- Abdulaziz Alqahtani performed the experiments, authored or reviewed drafts of the article, and approved the final draft.

### Ethics

The following information was supplied relating to ethical approvals (i.e., approving body and any reference numbers):

The study was conducted in the College of Dentistry Research Center (CDRC) at King Saud University's College of Dentistry in Riyadh, Saudi Arabia. Ethical permission was given by the CDRC (# LR 0513).

### Data Availability

The raw measurements are available in the Supplemental File.

### Supplemental Information

Supplemental information for this article can be found online at http://dx.doi.org/10.7717/peerj.20480#supplemental-information.

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
