# Peer review of "Bond strength between layering ceramic and zirconia core: influence of various surface conditioning of zirconia"

_PeerJ, doi:10.7717/peerj.20480_

## Round 0.1 · original submission · Major Revisions

· Academic Editor

Major Revisions

**Language Note:** When preparing your next revision, please ensure that your manuscript is reviewed either by a colleague who is proficient in English and familiar with the subject matter, or by a professional editing service. PeerJ offers language editing services; if you are interested, you may contact us at [email protected] for pricing details. Kindly include your manuscript number and title in your inquiry. – PeerJ Staff

Reviewer 1 ·

Basic reporting

1. More fluent and professional terminology must be used throughout the article, especially in the abstract. The current phrasing lacks precision and polish, which affects the overall readability and impact of the work.

2. The abstract requires additional detail in the methodology section. Specifically, the ceramic layering criteria over the zirconium core must be clearly described, including relevant dimensions and procedural specifics.

3. The introduction section provides general background knowledge, but most of the references cited are outdated. The authors must revise this section to include more recent and relevant literature. Additionally, in line 49, the use of the term “novel” is inappropriate. Veneering ceramics and ceramic restorations have been in use for over 25 years. The authors should instead refer to the current advancements, such as translucent, ultra-translucent, and super-translucent zirconium blocks.

Experimental design

The research question is clearly stated. However, the study design is not described, and no sample size calculation is provided to validate the grouping of the study. This omission must be addressed to ensure the scientific rigor of the methodology.

No sample size estimation is included, nor is there any clarification regarding the number of specimens used in each group. The authors must provide a justification for the grouping and include a formal sample size calculation.

In line 111, the manual veneering of ceramic is described as being performed entirely free-handed. This raises concerns about dimensional control. It is recommended that a mold cavity with predefined dimensions be fabricated to standardize the ceramic buildup process and ensure consistency across samples.

In lines 114–117, the authors must specify the equivalent intraoral service time that corresponds to the chosen thermal aging duration. This information is critical for contextualizing the experimental conditions and must be supported by a relevant reference.

In line 120, the parameters used for shear bond strength (SBS) testing are not detailed. The authors must indicate the testing speed (e.g., 1 mm/min) and support this with an appropriate reference to ensure methodological transparency.

Validity of the findings

The data presented in the study appear promising. However, additional testing is required to strengthen the scientific impact and reliability of the findings. Specifically, surface roughness measurements should be conducted following the different surface treatment methods. This will provide a more comprehensive understanding of the effects of each treatment and enhance the technical depth of the study.

Additional comments

Results require a more detailed description and analysis of the SEM used to assess the failure mode. The current explanation lacks depth and must be expanded to clarify how SEM findings support the conclusions drawn.

Testing surface roughness is essential and should be included. It will provide more comprehensive insight into the effects of different surface treatment methods and help interpret the bond strength values more accurately. Relying solely on SBS testing is insufficient for a complete evaluation.

The article requires careful revision to enhance its scientific and clinical contribution. Improvements in methodological detail, analytical depth, and clarity are necessary to meet publication standards.

Reviewer 2 ·

Basic reporting

Introduction
Comment 1: Please review the references.

Comment 2: The scientific gap the study aims to address is not clearly stated. Although the comparison between surface treatments on bond strength is mentioned, there is no clear statement about the scarcity of comparative data or controversies in the literature regarding these methods. It would be helpful to add a justification for the study: Are there similar studies? What makes this study important?

Comment 3: Improve the text flow. For example:
"The term 'esthetic' has evolved over the years for both professionals and patients. Porcelain-fused-to-metal (PFM) crown restorations are being replaced with more aesthetically acceptable restorations, namely all-ceramic crowns..."
The first sentence seems disconnected when the author continues discussing PFM. Enhance the transition between concepts—perhaps show that the demand for esthetics influenced the transition from PFM to all-ceramic crowns to create a more logical flow.

Comment 4: Review the English.
"Zirconia is the preferred substructure most of the time, nevertheless, because of its exceptional mechanical qualities, which are unaffected by the size or restrictions of the restoration site. Furthermore, zirconia-based materials are among the most dependable choices..."
The transition between these sentences is mechanical and disconnected. The use of “nevertheless” is inappropriate here, as the second sentence does not contradict the first.

Comment 5: Check line 51 for repeated authors.

Comment 6: Your objective is clear, but it would be beneficial to link it explicitly to clinical impact.

Experimental design

Methods

Comment 1: The section is functional but could be better organized with clear subheadings such as: Ethical Approval, Specimen Preparation, Surface Treatments, Veneering and Aging, Bond Strength Testing, SEM Analysis, and Statistical Analysis.

Comment 2: To ensure reproducibility, it is necessary to clarify the sintering process of the ceramic and the surface treatment protocols, which are only briefly mentioned. For example, specify Al₂O₃ sandblasting pressure, time, and distance; duration and method of hydrofluoric acid application (contact time, rinsing, etc.); brand or manufacturer of Zirface, etc.

Comment 3: It is unclear how the ceramic was applied: Was it molded with a matrix? Applied with a brush? Layered and fired incrementally? Adding that the process followed manufacturer guidelines would help.

Comment 4: Why were 6000 thermocycling cycles chosen? Please add the clinical time equivalent of this number.

Comment 5: Was a normality test performed before statistical analysis?

Validity of the findings

Results

Comment 1: The results section repeats in text what is available in tables. Describe only the relevant findings and statistically significant comparisons; refer the reader to tables for detailed data.

Comment 2: Watch for grammatical errors.

Comment 3:
"...Zirface group exhibited the highest mean SBS value of 20.05 MPa..." (line 192)
But the previously reported mean for Zirface is 26.56 MPa.
20.05 MPa appears to be the overall mean value, as shown in Table 2. Please verify these data.

Additional comments

Discussion

Comment 1: Although SEM is mentioned, no direct SEM results are reported in the Results section. The methodology does not specify the SEM equipment used.
“SEM provided valuable insight into the morphological changes... and their correlation with SBS performance.”

Comment 2: Improve the discussion of study limitations by specifying them clearly.

Comment 3: Explain why Zirface showed superior performance. Is it due to silane presence, reactive layer thickness, or its chemical action? The discussion should link results with known adhesive mechanisms.

Comment 4: The comparison with previous studies lacks depth; merely citing authors without explaining context or methods weakens the argument. For example,
“Our results agree with those of Farah et al. and Yang et al…”
Explain why the results are similar. What type of sandblasting was used in the cited studies? Was a primer applied? Were the aging cycles comparable?

Comment 5: You mention that the Zirface group had more cohesive failures, but do not explore the clinical significance of this finding.

Comment 6: The discussion ends by noting the best results but does not connect them to clear practical recommendations. Suggest a possible clinical recommendation based on the results, e.g.,
“Zirface may be preferred in clinical scenarios where intraoral conditioning is challenging.”

Conclusion
The conclusion largely repeats the results verbatim without providing an interpretative synthesis or broader clinical/scientific implications. It lists findings as isolated points without integrating them to highlight which method is preferable, considering risks, practicality, or long-term stability. The current list format does not promote clear argumentation. It is recommended to rewrite in paragraphs, synthesizing key findings, emphasizing clinical implications, and acknowledging limitations.

---

## Round 0.2 · Minor Revisions

· Academic Editor

Minor Revisions

Dear authors,

Please proceed to handle the minor remarks pointed out by the reviewer. Many thanks.

Reviewer 1 ·

Basic reporting

-

Experimental design

-

Validity of the findings

-

Additional comments

I would like to extend my appreciation for your commitment and efforts in revising and enhancing the quality of your research. Please find below a few comments and suggestions intended to further improve the clarity and scientific rigor of your manuscript:

1. Line 42: Kindly specify the type of study clearly (e.g., in vitro study).

2. Line 49: Please clarify the shape of the specimen. The dimensions provided seem more applicable to a bar-shaped sample rather than a circular one. As illustrated in Figure 2, the reference to “diameter” should be removed if the specimen does not have a circular cross-section.

3. Line 153: A null hypothesis has not been stated. Please include one to align with the standard scientific structure.

4. Line 172: The method of specimen fabrication requires more detail. Please specify whether an STL file was sent directly to the milling machine. Additionally, include the sintering parameters and expand on the specimen cleaning protocol, as using alcohol alone is insufficient.

5. Line 229: The statement that 5000 thermal cycles represent 6 months of clinical service should be revised. It is generally accepted that 6000 cycles simulate approximately 5 years of intraoral service.

6. Please support your conclusion that hydrofluoric acid (HF) treatment outperformed airborne-particle abrasion with relevant literature and provide more robust data to substantiate this claim.

7. Conclusion Section: Kindly limit this section to stating the key findings of your study. Detailed explanations and reasoning should be confined to the discussion section.

Thank you again for your valuable contribution.

Reviewer 2 ·

Basic reporting

The authors have adequately addressed all reviewer comments and provided the necessary clarifications and revisions throughout the manuscript. The responses are clear and comprehensive, and the revised version reflects significant improvement in methodological detail, organization, and scientific clarity

Experimental design

The authors have adequately addressed all reviewer comments and provided the necessary clarifications and revisions throughout the manuscript. The responses are clear and comprehensive, and the revised version reflects significant improvement in methodological detail, organization, and scientific clarity

Validity of the findings

The authors have adequately addressed all reviewer comments and provided the necessary clarifications and revisions throughout the manuscript. The responses are clear and comprehensive, and the revised version reflects significant improvement in methodological detail, organization, and scientific clarity.

Additional comments

The authors have adequately addressed all reviewer comments and provided the necessary clarifications and revisions throughout the manuscript. The responses are clear and comprehensive, and the revised version reflects significant improvement in methodological detail, organization, and scientific clarity.

---

## Round 0.3 · accepted · Accept

· Academic Editor

Accept

Dear authors,

All isues seem now resolved. I am accepting your manuscript for publication. Many thanks and congratulations.